# Faithfulness and the Notion of Adversarial Sensitivity in NLP Explanations

**Supriya Manna** and **Niladri Sett**[*]
SRM University AP, India
reachsmanna@gmail.com, settniladri@gmail.com

## Abstract

Faithfulness is a critical metric to assess the reliability of explainable AI. In NLP, current methods for faithfulness evaluation are fraught with discrepancies and biases, often failing to capture the true reasoning of models. We introduce *Adversarial Sensitivity* as a novel approach to faithfulness evaluation, focusing on the explainer's response when the model is under adversarial attack. Our method accounts for the faithfulness of explainers by capturing sensitivity to adversarial input changes. This work addresses significant limitations in existing evaluation techniques, and furthermore, quantifies faithfulness from a crucial yet under-explored paradigm.

## 1 Introduction

Deep learning-based Language Models (LMs) are increasingly used in high-stakes Natural Language Processing (NLP) tasks (Minaee et al., 2021; Samant et al., 2022). However, these models are extremely opaque. To build user trust in these models' decisions, various post-hoc explanation methods (Madsen et al., 2022) have been proposed (Jacovi et al., 2021). Despite their popularity, these explainers are frequently criticized for their 'faithfulness', which is loosely defined as how well the explainer reflects the underlying reasoning of the model (Lyu et al., 2024; Jacovi and Goldberg, 2020). In the context of NLP, explainers assign weights to each token indicating their importance in prediction, and faithfulness is measured by how consistent these assignments are with the model's reasoning. However, since the explainer is not the model itself (Rudin, 2019), practitioners have developed several heuristics to measure the quality of these assignments (DeYoung et al., 2019; Zhou et al., 2022a; Nguyen, 2018; Jain and Wallace, 2019; Hooker et al., 2019; Lyu et al., 2024).

A common assumption behind many of these heuristics is the *linearity* assumption, which posits that the importance of each token is independent of the others (Jacovi and Goldberg, 2020). Based on this, a group of practitioners hypothesised that removing important tokens indicated by a faithful explainer should change the prediction, whereas removing the least important ones should not. Jacovi et al. (Jacovi and Goldberg, 2020) addressed these as *erasure*. DeYoung et al. (DeYoung et al., 2019) generalize the same with comprehensiveness and sufficiency. However, it has been exhaustively shown that the removal of features can produce counterfactual inputs[1] that are out of distribution (Hase et al., 2021; Chrysostomou and Aletras, 2022; Lyu et al., 2024; Janzing et al., 2020; Haug et al., 2021; Chang et al., 2018), socially misaligned (Jacovi and Goldberg, 2021), and often severely *pathogenic* (Feng et al., 2018). Furthermore, evaluation metrics such as *Area Under the Perturbation Curve* (AUPC) (Samek et al., 2016) are suspected to be severely *misinformative* (Ju et al., 2021). Instead of evaluating faithfulness, these methods primarily compute the similarity between the evaluation metric and explanation techniques, assuming the evaluation metric itself to be the ground truth (Ju et al., 2021).

Another line of work, known as adversarial robustness (Baniecki and Biecek, 2024), assumes that similar inputs with similar outputs should yield similar explanations. However, Ju et al.(Ju et al., 2021) has empirically shown that the change in attribution scores may be because the model's reasoning process has genuinely changed, rather than because the attribution method is unreliable. Moreover, this assumption is mainly valid when the model is 'astute'

---

[*]Corresponding author

[1]Counterfactual inputs (CI) & counterfactual explanations (CE) are completely different. Removing features from the main input makes CI wrt the actual input. Miller et al. used this terminology (Miller, 2019). We've discussed CE in Section 6. Hase et al. (Hase et al., 2021) debunked the same confusion of the reviewers here.

(Bhattacharjee and Chaudhuri, 2020; Khan et al., 2024) and doesn't necessarily apply to explainers that don't perform local function approximation for feature importance estimation (Han et al., 2022). As a result, this assumption is practically *restrictive* and *vague*, leading practitioners to hesitate in endorsing this approach for assessing faithfulness (Lyu et al., 2024).

Across almost all popular lines of thought, the settings in which faithfulness is quantified are *linear* (Jacovi and Goldberg, 2020), *restrictive* (Khan et al., 2024), *misinformative* (Ju et al., 2021), and thus the judgements on explainer quality based on such quantification could be arguable. Since, understanding the model's reasoning is challenging, and aforementioned assumptions are often deceptive, in this work, we take a fundamental approach. Previous research has demonstrated that deep models are not only opaque but also severely fragile (Goodfellow et al., 2014; Szegedy et al., 2013). As explainers are primarily to facilitate *trust* on these complex models, we argue that a faithful explainer is obligated to uncover such vulnerabilities and anomalous behaviour of the model to the end user. In this context, we introduce the notion of 'adversarial sensitivity' for the explainers. We seek the most similar (semantically and/or visually) counterpart(s) from the entire input space (subjected to certain constraints) that produces a different output, aka 'adversarial examples' (Goodfellow et al., 2014). These pairs of inputs are always bounded by a certain distance, ensuring they are sufficiently comparable. Consequently, unlike counterfactuals, these pairs are much less likely affected by abrupt *semantic shifts* (Lang et al., 2023) that often lead to out-of-distribution scenarios (Hendrycks and Gimpel, 2016; Sun and Li, 2022; Sun et al., 2021; Liang et al., 2017), making our comparisons more nuanced and robust. However, since these pairs yield different outputs, their underlying reasoning in the model is bound to differ (Jacovi and Goldberg, 2020; Adebayo et al., 2018). Faithful explanations should reflect these changes, highlighting the difference in the model's inherent reasoning. We formally define the same as 'adversarial sensitivity' of the explainers. Our contributions in this paper are summarised as:

- we introduce the notion of 'adversarial sensitivity' of an explainer, and propose a necessary test for faithfulness based on it;

- we present a robust experimental framework

to conduct the faithfulness test;

- we conduct the proposed faithfulness test on six state-of-the-art post-hoc explainers over three text classification datasets, and report its (in)consistency with popular erasure based tests.

This paper is organised as follows: We introduce the notion of adversarial sensitivity, exploring its significance and relation with faithfulness in Section 2. In Section 3, we details our methodology, outlining the framework used to conduct our investigations. In Section 5, we present our findings, offering in-depth analysis and interpretations of the data. We contextualize our work within the broader research landscape in Section 6, highlighting how our study contributes to and extends existing knowledge. Finally, in Section 7, we conclude by summarizing our key findings and proposing directions for future research, emphasizing the potential avenues for further exploration.

## 2 Adversarial Sensitivity

In this section, we introduce the notion of adversarial sensitivity, exploring its significance and relation with faithfulness. Thereafter, we propose the guideline for evaluating faithfulness with adversarial sensitivity.

**Definition 1.** *Adversarial Example* (AE): Given a model $f : X \rightarrow Y$, where $X$ is the space of textual inputs and $Y$ is the set of classes, if there exists $x'$ for a given input $x \in X$ such that:

$$\{x' \in X \mid S(x, x') \geq \theta \text{ and } f(x) \neq f(x')\},$$

we call $x'$ an *adversarial example* (AE), where $S(\cdot, \cdot)$ is a similarity measure and $\theta$ is a predefined similarity threshold.

**Definition 2.** *Local Explanation*: A local feature importance function $I$ takes an instance $x \in X$ and the model $f$ as input, and produces a weight vector as output:

$$I(f, x) = W_{x,f} = (w_1, w_2, \ldots, w_n),$$

where $w_i$ represents the *importance* of the $i$-th token $x_i$ for the prediction $f(x)$.

**Definition 3.** *Adversarial Sensitivity*: Adversarial Sensitivity for a local explainer $I$ for $(x, x')$ is given by $d(W_{x,f}, W_{x',f})$. Here, $x'$ is an AE of input $x$, and $d(\cdot, \cdot)^2$ is a distance measure.

---

[2]in details at Section 3

**Adversarial Sensitivity and Faithfulness**: Given $x'$ is an AE of $x$ for $f$, if $I$ is 'faithful' to $f$, then $W_{x,f}$ and $W_{x',f}$ should be dissimilar. In our setup, we report the mean distance over all obtained pairs of $(x, x')$. This is a necessary but not sufficient condition for faithfulness. Currently (at the time of writing this paper) there is no necessary and sufficient condition for faithfulness (Lyu et al., 2024). However, following the argument of Lyu et al. (Lyu et al., 2024), as these metrics are primarily (meta)heuristic based evaluations, accessing faithfulness with several necessary tests is much more practical than attempting to formulate an exhaustive list of necessary and sufficient conditions and then evaluating against all of them. Adversarial Sensitivity is one of such necessary tests to evaluate the *faithfulness* of explainers.

Adversarial Machine Learning research has extensively demonstrated that even minimal perturbations in the input space can deceive well-trained models (Alzantot et al., 2018; Garg and Ramakrishnan, 2020; Li et al., 2020; Gao et al., 2018; Ebrahimi et al., 2017; Kuleshov et al., 2018; Zang et al., 2019; Pruthi et al., 2019; Jin et al., 2020; Li et al., 2018; Ren et al., 2019). Given the discrete and combinatorially large nature of the input space, finding all possible adversarial examples (AEs) under all possible constraints is often impractical, especially in a black-box setting. Therefore, we advocate for greedily searching for AEs within a well-tested set of constraints to avoid obfuscating and low-quality examples. In this study, we select extensively used word-level, character-level, and behavioural invariance constraints. Whether methods like back-translation, paraphrasing, or hybrid attacks etc (Zhang et al., 2020) maintain semantic and structural similarity while generating AEs, and suitability for faithfulness evaluation are kept for further study.

Obtaining AEs is conducted in two ways: assuming the model to be either white-box or black-box. In a white-box setting, gradients are primarily used first to identify the importance of tokens and then perturb them to create an adversarial input if the output changes. For our setting, this approach has two distinct problems. Firstly, gradient-based feature importance can be untrustworthy and manipulative (Wang et al., 2020; Feng et al., 2018). Secondly, a class of post-hoc explainers (e.g., Gradient, Integrated Gradient) also uses the gradient to retrieve the importance of tokens. Comparing these with explainers that do not use gradient informa-

tion, such as LIME or SHAP, may lead to biased comparisons. Lastly, popular gradient-based attacks such as HotFlip (Ebrahimi et al., 2017) are often less likely to adhere to perturbation constraints while crafting adversarial examples (Wang et al., 2020). Therefore, we do not consider investigation on white-box attacks for *adversarial sensitivity* and adhere to a more practical, model-agnostic, and transferable black-box attacking framework. However, even in the black-box settings we employ some ad-hoc heuristics for greedily perturbing the words based on its relative importance (Zhang et al., 2020), but modern explainers do not use such ad-hoc methods for calculating feature importance (Lyu et al., 2024; Madsen et al., 2022). Therefore, our faithfulness test is unbiased towards the underlying mechanisms of (almost) all types of modern post-hoc explainers.

## 3 Faithfulness Test Setup

### 3.1 Obtaining AEs

Primarily, obtaining AEs (an *adversarial attack* on the model) is a greedy or brute-force procedure, where a search algorithm iteratively selects locally optimal constrained perturbations until the label changes (Morris et al., 2020). As mentioned in Section 2, we devise our attacks in three constraint classes: word level, character level, and behaviorul invarince. We brief the implementation details of these attacks as follows.

### 3.1.1 Word Level (*A1*)

We adhere to the constraints proposed in the strong baseline 'TextFooler' (Jin et al., 2020) while implementing our word-level attack (*A1*). Initially, we assign weights to each word based on its impact on the model's prediction when removed. Then, in decreasing order of importance, we take each word (except stopwords), find semantically and grammatically correct $K$ (we set $K = 50$) words to replace the selected word, and generate all possible intermediate corpus and query the model. If the best result (which alters the prediction the most) from this pool exceeds the one from the previous iteration, we select the new one as the current result; otherwise, we stick to the previous one. This process iterates until the current result yields a different output or we have exhaustively searched the set of possible results and found none that alter the output.

Although the constraints, including vocabulary se-

lection and stopwords filtering, were effective in crafting adversarial examples, we observed some discrepancies with off-the-shelf hyperparameter selections. Consequently, we adjusted the minimum word embedding cosine similarity to 0.5 (instead of 0.7) and set an angular similarity threshold of 0.84 within a 15-token window.[3]

### 3.1.2 Character Level (*A2*)

For character-level attack (*A2*), we assign weights to each word based on its impact on the model's prediction when replaced with an unknown token ('[UNK]'). The rest of the procedure is the same as the *A1*, but instead of semantically similar words, we replace the selected word after applying a combination of character-level perturbations proposed by Gao et al. (Gao et al., 2018), subject to a pre-defined edit distance threshold, proposed in (Gao et al., 2018). Li et al. (Li et al., 2018) empirically showed that character-level perturbation can change semantic alignment in the embedding space. Therefore, after filtering with edit distance, we also employ the universal sentence encoder (Cer et al., 2018) and use the similarity threshold proposed in (Li et al., 2018) to select the final candidate.

### 3.1.3 Behaviorul Invarience (*A3*)

Recently, Ribeiro et al. (Ribeiro et al., 2020) emphasised that models are hypersensitive not only to minute perturbations but also to 'invariant' tokens. Ribeiro et al. proposed 'Checklist' that evaluates models across diverse linguistic capabilities such as vocabulary, syntax, semantics, and pragmatics. For our setting, we adopt the 'Invariance Testing' they proposed (*A3*). We change names, locations, numbers, etc., wherever feasible in the sentences and check if these alterations affect the prediction. As Ribeiro et al. (Ribeiro et al., 2020) showed, a model should not be sensitive to such parameters. If it is, it indicates an inability to handle commonly used linguistic phenomena, which are subsequently characterised as a type of adversarial example (Morris et al., 2020). We employ the off-the-shelf implementation of the invariance testing from 'TextAttack' (Morris et al., 2020). In our datasets, we do not have a lot of instances where phone numbers, locations, age etc are present and as we are changing these only once in this attack

(else it could lead to an infinite loop), the success rate of this attack is lesser than the other attacks. However, from a linguistic perspective, this attack is crucial to make our experiments exhaustive.

In all these attacks, we do not perturb stop-words. Next, we only consider the example as successful AE if the prediction confidence crosses a certain threshold (we set it to be at least 70%). Finally, as we are conducting model-agnostic attacks, we acknowledge that even if the constraints are reasonably restrictive, there is always a chance that any of these examples could be out-of-distribution (OOD). To mitigate such issues, we follow a robust baseline wherever required for detecting OOD scenarios by computing the 'maximum/predicted class probability' (MCP) from a softmax distribution for the predicted class of each AE (Hendrycks and Gimpel, 2016). MCP has been evaluated as a strong baseline, particularly when the underlying model is fine-tuned (e.g., BERT, RoBERTa) (Hendrycks et al., 2020; Desai and Durrett, 2020). We empirically selected only those adversarial examples that had a probability exceeding 70% across all attacks and datasets.

## 3.2 Measuring the Distance

To measure the dissimilarity of the explanations, we follow the distance measure given by Ivankay et al. (Ivankay et al., 2022), that is:

$$d = 1 - \frac{\tau(W_{x,f}, W_{x',f}) + 1}{2} \qquad (1)$$

where $\tau(\cdot, \cdot)$ is a correlation measure. Ivankay et al. (Ivankay et al., 2022) chose Pearson correlation for their distance measure. But while creating adversarial examples, a common phenomenon is obtaining unequal token vectors for $(x, x')$ due to tokenisation (Sinha et al., 2021). Correlation measures like Pearson, Kendall, and Spearman cannot handle disjoint and unequal ranked lists. Sinha et al. (Sinha et al., 2021) used heuristics like Location of Mass (LOM) (Ghorbani et al., 2019) to mitigate such issues. But Burger et al. (Burger et al., 2023) highlighted their shortcomings and employed Rank Based Overlap (RBO) (Webber et al., 2010) metric. While RBO may be robust, it introduces complications, particularly with its selection of free parameter '$p$' determining the *user persistence*.[4]

---

[3]We discovered that the authors of TextAttack (Morris et al., 2020) identified bugs in the original implementation of TextFooler (Jin et al., 2020) and suggested a set of hyperparameters that were mostly coherent in our setup. Details can be found here.

[4]Burger et al. (Burger et al., 2023) used LIME's feature importance along with explanation's average length to determine the value of '$p$' for their experimentation and Goren et al. (Goren et al., 2018) apparently used an ad-hoc value of $p = .7$ in their experimental setup.

Moreover, the assumption on the *depth* in RBO using Bernoulli's random variable and *weights* of overlaps in explanation using geometric distribution may not be always adequate as per our setting. Furthermore, the selection between the base and extrapolated versions of RBO gives rise to the disparity in 'sensitivity', especially when the *residual* is significant (Webber et al., 2010). Following the arguments of Jacovi et al. (Jacovi and Goldberg, 2020) we, do not endorse unnecessary human intervention in faithfulness studies. As RBO inherently carries the notion of the persistence of *users*, we didn't select RBO for this work.

We have extensively investigated selecting the similarity measures in previous works, but none of the works has tackled the problem of unequal and/or disjoint rank lists from an axiomatic perspective that will be adequate for our setting. Emond et al. (Emond and Mason, 2002) proposed a new correlation coefficient designed to accommodate incomplete and non-strict rankings; however, this metric is not considered due to the lack of formal proof or empirical evidence. Later, Monero et al. (Moreno-Centeno and Escobedo, 2016) introduced essential axioms for a distance measure between incomplete rankings, establishing the existence and uniqueness of such a measure and demonstrating its superiority in generating intuitive consensus rankings compared to alternative methods. Following these axioms, we adopt the nonparametric correlation coefficient '$\hat{\tau}_x$' presented in Yoo et al. (Yoo et al., 2020), which highlights the inadequacy of the $\tau_x$ ranking correlation coefficient devised in (Emond and Mason, 2002) in ensuring a neutral treatment of incomplete rankings. Moreover, our employed non-parametric correlation coefficient '$\hat{\tau}_x$' is a generalization of Kendall $\tau$ on the aforementioned axiomatic foundation established by Monero et al. (Moreno-Centeno and Escobedo, 2016) for handling a variety of ranking inputs, including incomplete and non-strict ones. Therefore, $\hat{\tau}_x$ is foundationally much robust and can handle several types of tokenization discrepancies. Furthermore, this very distance is a nonparametric generalization of the kemeny-snell distance (Kemeny and Snell, 1962) for nonstrict, incomplete ranking space (Moreno-Centeno and Escobedo, 2016). As a result, unlike the previous distance metric, '$\hat{\tau}_x$' is not only robust but also enjoys the properties that the Kemeny-snell distance retains for all types of rankings produced by the tokenisers.

## 3.3 Interpreting the Results

Our proposed test is a necessary test for faithfulness based on the desideratum that the explainers should produce *different* explanations for AEs. Obtaining AEs is always subject to different sets of constraints. As a result, each attack type i.e. *A1, A2, A3* is disjoint in nature thus, each of them independently conducts a necessary test given they produce *successful* AEs. Theoretically, there can be finitely many AEs if we keep changing the set of constraints but in this paper, we followed three extensively evaluated, diverse sets of constraints to empirically demonstrate the adversarial sensitivity of explainers around these disjoint constraint sets. As a result, our setup consists of three disjoint necessary tests for inspecting faithfulness using adversarial sensitivity. We evaluate the explainers on the basis of how much sensitivity they obtain for how many number of discrete constraint sets. However, as these are all necessary tests, the primary objective is to reject the unfaithful ones. Also, it is highly seek-worthy that explainers perform *consistently* well across constraint sets. Now, if the results across constraint sets are fluctuating for a given setup, it could be confusing for the end user to evaluate the explainers holistically. This is why, for an aggregated ranking we recommend using a consensus aggregation (e.g., Kemeny-young aggregation (Kemeny, 1959)) over empirical evaluation. Although, in our experiments, we obtained consistent results across *A1, A2, A3*.

## 4 Experimental Setup

### 4.1 Datasets and Models

We conducted our experiments on SST-2 (Socher et al., 2013), and Tweet-Eval (Hate) (Barbieri et al., 2020) for binary classification, and on AG News (Zhang et al., 2015) for multi-class classification. We fine-tuned a Distill BERT and a BERT-based model (Devlin et al., 2018) until it achieved a certain level of accuracy for each dataset, and attacked it with the three attack methods *A1, A2, and A3* described in the Section 3.1. We report the models' accuracy before and after each attack[5] in Table 1. We've addressed 'Tweet-Eval (Hate)' as 'Twitter' throughout the paper and Distill BERT-based model (Sanh, 2019) as DERT in Table 1. We used the standard train, test split for each dataset from the huggingface library and reported results

---

[5]If AE is not obtained, the attack is failed and vice-versa.

up to the second decimal place.

## 4.2 Explainers and Faithfulness Metrics Details

Commonly used post-hoc local explainers can be broadly categorised in two types: perturbation-based and gradient-based explainers (Madsen et al., 2022). We have considered two commonly used perturbation-based model agnostic explainers: LIME (*LIME*) (Ribeiro et al., 2016) and SHAP (*SHAP*) (Lundberg and Lee, 2017). For *SHAP*, we use the default selection of partition shap. [6] From gradient based ones, we have chosen Gradient (*Grad.*) (Simonyan et al., 2013), Integrated Gradient (*Int. Grad.*) (Sundararajan et al., 2017) and their *xInput* version: Gradient × Input (*Grad. × Input*), and Integrated Gradient × Input (*Int. Grad. × Input*). We compare our findings with extensively used erasure (Jacovi and Goldberg, 2020) based metrics: comprehensiveness, sufficiency (DeYoung et al., 2019), and correlation with 'Leave-One-Out' scores (Jain and Wallace, 2019) for faithfulness comparison. The Appendix contains the description of erasure-based faithfulness metrics and post hoc explainers used in our experiments.

We run our experiments on an NVIDIA DGX workstation, leveraging Tesla V100 32GB GPUs. We use *ferret* with default (hyper)parameter selection (Attanasio et al., 2022) for both erasure metrics and explanation methods, *TextAttack* (Morris et al., 2020), *universal sentence encoder* (Cer et al., 2018) across attacking mechanism. We wrote all experiments in Python 3.10. Our total computational time to execute all experiments is roughly 18 hours. We report the consolidated findings for both models below in Table 2.

## 5 Results & Discussion

From Table 2, it is clearly observable that as per Adv. Sens., LIME, SHAP, Gradient × Input, and Integrated Gradient × Input all perform competitively across various datasets and attacks. However, the vanilla versions of gradient-based methods are not as effective. Notably, the Gradient itself exhibits the least sensitivity to adversarial inputs, followed by Integrated Gradient. Furthermore, Integrated Gradient's adv. sens. remains almost invariant to the type of attacks across all datasets, unlike comp. and suff. Interestingly, all explainers except Gradient show a drop in sensitivity in

the AG News dataset across all attacks. Gradient performs best on all attacks in AG News amongst datasets. Perturbation-based explainers like LIME and SHAP are among the best performers across datasets. Gradient × Input and Integrated Gradient × Input perform well within the group of white-box explainers, with LIME and SHAP.

Under erasure methods across all datasets, Gradient is a moderately well-performing explainer, whereas Gradient × Input performs much worse. However, according to Adversarial Sensitivity, Gradient × Input is one of the best performers, with Gradient being the worst among all. Like Gradient × Input, Integrated Gradient also largely performs worse than Gradient in erasure, but it remains consistently moderate according to Adv. Sens. Both LIME and SHAP not only perform very well in both Adv. Sens. and erasure metrics but also the difference b/w their magnitudes for both erasure metrics and adv. sens. are (considerably) nominal. Integrated Gradient × Input is substantially similar to LIME, SHAP in adv sens., but we observe a considerable drop in comprehensiveness for SST-2 and AG News for both the models, unlike adv. sens.

To demonstrate, how to evaluate the explainers based on the consensus ranking, we are considering the case of SST-2 for the BERT Model. We use the Kemeny-Young method here; as this has been extensively used for Condorcet ranking (Young, 1988); it also satisfies highly desirable social choice properties for *fair* voting (Owen and Grofman, 1986; Young, 1995). Kemeny-Young aggregation also have been used in biology and social science extensively (Brancotte et al., 2014; Andrieu et al., 2021; Arrow et al., 2010). We first convert the columns of *A1, A2, A3* into ranking vectors using a ranking function. In our case, we used the traditional ranking: the higher the score (here the *score* is average distance obtained), the lower the ranking. We obtained the consensus ranking vector as $[2, 3, 6, 5, 4, 1]$. Here, the indices of the vector denote the respective position of explainers (starting from 1 onwards) in the '**Explainer**' column.

DeYoung et al. (DeYoung et al., 2019) advocated for both **high** comprehensiveness and **low** sufficiency for adequate explanations but unlike us; they did not propose any consensus evaluation for explainers with these two parameters taken together. According to the definition, both metrics measure two different aspects of explanations. This makes the evaluation of explainers even confusing

---
[6]partition shap documentation:

Table 1: Accuracy, before and after attacks – Distill BERT and BERT

| Model | Dataset | Accuracy (%) | Accuracy after *A1* (%) | Accuracy after *A2* (%) | Accuracy after *A3* (%) |
|---|---|---|---|---|---|
| DERT | *SST-2* | 91.50 | 8.42 | 21.61 | 99.62 |
| | *AG News* | 93.10 | 26.71 | 68.91 | 91.46 |
| | *Twitter* | 51.7 | 18.84 | 8.28 | 96.93 |
| BERT | *SST-2* | 92.43 | 10.77 | 19.00 | 99.42 |
| | *AG News* | 94.40 | 25.00 | 32.00 | 94.50 |
| | *Twitter* | 54.32 | 23.58 | 12.61 | 95.29 |

Table 2: Consolidated Findings

| Model | Explainer | SST-2 Erasure Comp. ↑ | Suff. ↓ | LOO ↑ | SST-2 Adv. Sens. ↑ A1 | A2 | A3 | AG News Erasure Comp. ↑ | Suff. ↓ | LOO ↑ | AG News Adv. Sens. ↑ A1 | A2 | A3 | Twitter Erasure Comp. ↑ | Suff. ↓ | LOO ↑ | Twitter Adv. Sens. ↑ A1 | A2 | A3 |
|---|---|---|---|---|---|---|---|---|---|---|---|---|---|---|---|---|---|---|---|
| DERT | *LIME* | 0.72 | 0.02 | 0.32 | 0.77 | 0.72 | 0.81 | 0.68 | -0.03 | 0.21 | 0.66 | 0.64 | 0.72 | 0.89 | 0.00 | 0.37 | 0.77 | 0.75 | 0.83 |
| | *SHAP* | 0.70 | 0.02 | 0.27 | 0.76 | 0.74 | 0.80 | 0.63 | -0.03 | 0.13 | 0.64 | 0.61 | 0.70 | 0.85 | 0.00 | 0.33 | 0.76 | 0.73 | 0.84 |
| | *Grad.* | 0.37 | 0.10 | 0.10 | 0.18 | 0.2 | 0.07 | 0.44 | 0.03 | 0.06 | 0.21 | 0.23 | 0.13 | 0.76 | 0.07 | 0.13 | 0.18 | 0.2 | 0.09 |
| | *Int. Grad.* | 0.20 | 0.32 | -0.04 | 0.56 | 0.55 | 0.55 | 0.03 | 0.27 | -0.04 | 0.52 | 0.53 | 0.54 | 0.26 | 0.50 | -0.03 | 0.56 | 0.55 | 0.58 |
| | *Grad. x Input* | 0.17 | 0.35 | -0.12 | 0.71 | 0.63 | 0.83 | 0.04 | 0.23 | -0.11 | 0.59 | 0.57 | 0.69 | 0.29 | 0.43 | -0.10 | 0.71 | 0.67 | 0.82 |
| | *Int. Grad. x Input* | 0.53 | 0.08 | 0.24 | 0.76 | 0.70 | 0.80 | 0.54 | 0.00 | 0.12 | 0.58 | 0.57 | 0.54 | 0.81 | 0.02 | 0.22 | 0.76 | 0.72 | 0.84 |
| BERT | *LIME* | 0.68 | 0.01 | 0.33 | 0.74 | 0.75 | 0.86 | 0.72 | -0.06 | 0.14 | 0.64 | 0.54 | 0.68 | 0.86 | 0.00 | 0.32 | 0.76 | 0.75 | 0.82 |
| | *SHAP* | 0.61 | 0.02 | 0.26 | 0.71 | 0.70 | 0.84 | 0.67 | -0.05 | 0.11 | 0.62 | 0.52 | 0.67 | 0.87 | 0.01 | 0.35 | 0.80 | 0.79 | 0.86 |
| | *Grad.* | 0.36 | 0.09 | 0.10 | 0.17 | 0.18 | 0.04 | 0.51 | 0.04 | 0.03 | 0.23 | 0.34 | 0.14 | 0.78 | 0.05 | 0.16 | 0.16 | 0.17 | 0.07 |
| | *Int. Grad.* | 0.19 | 0.29 | 0.00 | 0.53 | 0.55 | 0.52 | 0.04 | 0.26 | -0.03 | 0.51 | 0.50 | 0.50 | 0.22 | 0.38 | -0.04 | 0.51 | 0.52 | 0.51 |
| | *Grad. x Input* | 0.22 | 0.27 | 0.01 | 0.66 | 0.67 | 0.86 | 0.46 | 0.06 | 0.16 | 0.62 | 0.54 | 0.67 | 0.21 | 0.41 | 0.00 | 0.73 | 0.71 | 0.71 |
| | *Int. Grad. x Input* | 0.54 | 0.06 | 0.02 | 0.76 | 0.76 | 0.85 | 0.47 | 0.04 | 0.05 | 0.56 | 0.52 | 0.56 | 0.83 | 0.01 | 0.18 | 0.75 | 0.75 | 0.74 |

with comprehensiveness-sufficiency, especially if the results for these two metrics are fluctuating. We did not find any axiomatically valid evaluation strategy for explainers in the presence of different kinds of faithfulness metrics in subsequent literature (including DeYoung's paper (DeYoung et al., 2019)) as well. It is worth noting Javoci et al. (Jacovi and Goldberg, 2020) reported the same observation previously. As Javoci et al. said, "*Lacking a standard definition, different works evaluate their methods by introducing tests to measure properties that they believe good interpretations should satisfy. Some of these tests measure aspects of faithfulness. These ad-hoc definitions are often unique to each paper and inconsistent with each other, making it hard to find commonalities.*" (Jacovi and Goldberg, 2020).

Although evaluation metrics are inherently different from one another, for the sake of demonstrating an inter-comparison between erasures and adv. sens.[7], we rank the explainers based on the *scores* they obtain in individual erasure methods in the case of SST-2 for the BERT Model in Table 2. We consider the same ranking function used for adv. sens. for Comprehensiveness and LOO score and the inverse of the same ranking function for Sufficiency due to its opposite nature with respect to the former. First, we take the Kemeny-Young aggregation of comprehensiveness and sufficiency; the

ranking obtained is $[1, 2, 4, 6, 5, 3]$. LOO's ranking is: $[1, 2, 3, 6, 5, 4]$. Next, we combine all erasure columns and get the aggregation as $[1, 2, 4, 6, 5, 3]$. The obtained aggregated ranking for adv. sens. was $[2, 3, 6, 5, 4, 1]$. From this comparison, we retrieve all explainers have obtained different rankings for comprehensiveness-sufficiency, LOO, and combined aggregation of erasures, as compared with adv sens. Throughout our experiments for both models, we observed explainers except for LIME & SHAP (as mentioned earlier) are largely inconsistent with one or more erasure method(s).

Nevertheless, erasure has been used in several novel affairs and benchmarkings (Mathew et al., 2021; Atanasova et al., 2023; Liu et al., 2022; Babiker et al., 2023) due to its easy-to-implement and seemingly reasonable assumption. However, we observe in our experimentation that erasure methods are inconsistent except perturbation based explainers with our proposed metric. Unlike erasure, which makes simplistic assumptions about the independence of the token's importance and absence of non-sensical OOD results while removing tokens (Lyu et al., 2024), adversarial sensitivity is founded on the assumption that faithful explainers should capture the intrinsic dissimilarity of model reasoning when *fooled*. We, therefore, advocate for the adoption of adversarial sensitivity as a foundational metric for a necessary test of faithfulness for assessing explainers.

---

[7]we do not necessarily endorse this rank-based comparison as an axiomatic comparison in the presence of different type of faithfulness evaluation parameters but a (hard) estimate in the absence of such comparisons.

## 6 Related Works

**Faithfulness evaluation**, based on previous literature, can be broadly categorised in six ways: axiomatic evaluation, predictive power evaluation, robustness evaluation, perturbation-based evaluation, white-box evaluation, and human perception evaluation (Lyu et al., 2024). The commonly used erasure is primarily a perturbation-based evaluation: it hypothesised that the change in model's output caused by the removal tokens is proportional to the *importance* of the tokens for the prediction. If a local explainer is *faithful*, removal of *important* tokens as identified by the explainer should align with the hypothesis. *Comp., Suff., LOO* are different instances of the erasure hypothesis. Our hypothesis is also somewhat related to the perturbation-based evaluation. We hypothesised that a *faithful* explainer should be sensitive to anomalous input that *fools* the model. We perturb at several levels in the input to deceive the model, not to interpret. Next, we evaluate how much the explainer is *sensitive* towards the subtle changes that deceive the model. As the deep models are known to be severely fragile, we argue this is a necessary quality for the explainer to be *faithful* when the model is not showing its *expected* behaviour.

Following the hypothesis of adversarial robustness, which comes under the robustness evaluation category, successful *adversarial attack* on explainer aims to perturb the input such that an explainer generates dissimilar (non-robust) explanations subject to 'similar' input and 'similar' (bounded by a *certain* distance) output (Baniecki and Biecek, 2024) (AdvxAI). However, rather than any ad-hoc distance to compare the similarity of explainer Alvarez et al. (Alvarez-Melis and Jaakkola, 2018) emphasize on the (local) lipschitz continuity measurement in this setting. Khan et al. (Khan et al., 2024) has recently analysed the theoretical bounds of (dis)similarity under this setting when the explainer and classifier (Bhattacharjee and Chaudhuri, 2020) are astute. Anyways, AdvxAI is not a formally accepted measure of faithfulness (Ju et al., 2021; Zhou et al., 2022b; Lyu et al., 2024), as the model may yield different reasoning rather than the explanation is non-robust. Anyhow, in this work we conduct attacks to deceive the model, not the explainer following the aforesaid hypothesis. For a broad overview on faithfulness evaluation we suggest the reader to refer to (Lyu et al., 2024).

**Adversarial examples** (AdvAI) can be crafted at several levels: word level, character level, phrase level, paraphrasing, back translation, invariance testing, etc. in white-box and black-box settings primarily (Zhang et al., 2020). We employed word level, char level and invarience testing attacks. Noppel et al.(Noppel and Wressnegger, 2023) systematised the underlying relations of AdvAI and AdvxAI. For a broder overview, we refer the reader to (Qiu et al., 2022; Zhang et al., 2020). Adversarial attacks on NLP systems have been carried out primarily in 2 types: white box and black box (Zhang et al., 2020), we didn't go with white box ones as they primarily leverage gradient information also, as several explainers such as Integrated Gradient or Gradient access the same information which could constitute a biased evaluation (Ju et al., 2021) as the attacking mechanism and the explanation method are similar and both leverage gradient information.

**Counterfactual explanations** (Mothilal et al., 2020), which demonstrate the changes would produce a distinct outcome, differ fundamentally from adversarial examples (Freiesleben, 2022), which aim to deceive models with minimal input changes. Counterfactuals should be semantically and/or visually different (Yang et al., 2020). Thus, it is not intended to deceive the underlying model. In the context of Natural Language Inference (NLI), Atanasova et al. (Atanasova et al., 2023) experimented with counterfactuals to investigate faithfulness. Camburu et al. (Camburu et al., 2019) explored inconsistencies in explanations for NLI but did not adhere to the constraints necessary for generating adversarial inputs required in our setting. Moreover, counterfactual explanations can potentially highlight necessary features but may miss sufficient ones for prediction (Hsieh et al., 2020).

**Similarity measures** in previous works, especially in AdvxAI (Sinha et al., 2021; Burger et al., 2023; Ivankay et al., 2022), have used mainly correlations, distance measures, and top 'k'% intersection in tokens. Burger et al. (Burger et al., 2023) comprehended the common issues with such metrics due to tokenization discrepancies and employed RBO (Webber et al., 2010). We did not select RBO having the free parameter *user persistence (p)*, as we argue that faithfulness should not be based on the *unnecessary* human evaluations. We rather select the distance invented by Moreno et al. (Moreno-Centeno and Escobedo, 2016) that satisfies all the axioms for non-strict, incomplete rank-

ings and also satisfies the desirable social choice properties of the Kemeny-snell distance (Kemeny and Snell, 1962) for *fair and conclusive* rankings.

## 7 Conclusion and Further Work

In this work, we explored the shortcomings of widely used faithfulness measures in NLP and proposed a test to evaluate explainers based on their sensitivity to adversarial inputs. Through extensive experiments on six post-hoc explainers, we found that gradient & integrated gradient aren't (sufficiently) sensitive, while LIME, SHAP, and Gradient × Input, and Integrated Gradient × Input show better sensitivity. We also observed notable differences between our evaluation and traditional erasure-based faithfulness measures.

Future work will explore adversarial sensitivity for multilingual datasets, low-resource languages, and advanced lms.

## Broader Impact

Deep models are not only fragile but also opaque. Our work lies at the intersection of these two critical aspects. Building on the arguments presented by Jacovi et al. (Jacovi and Goldberg, 2020), we introduce a necessary test for assessing faithfulness. Given that the underlying assumption of adversarial sensitivity is applicable to (nearly) all data types and models, this concept can be extended across (almost) all domains and explanation mechanisms.

Faithfulness is a key component in explainable AI (Miller, 2019). When a model behaves deceptively under any form of adversarial intervention, it becomes imperative that explainers provide *faithful* explanations in such scenarios, rather than merely those where the model performs according to user expectations. Adversarial sensitivity aids end-users in identifying explainers that are *responsive* to adversarial instances. We strongly believe that the nuanced notion of adversarial sensitivity opens up a new direction for evaluating explainers, particularly in situations where being *unfaithful* could lead to a misinterpretation of why the model produces deceptive results.

## Limitation

Adversarial attacks are computationally expensive. Our work therefore is much computationally expensive and non-trivial than erasures. Our work is a necessary test faithfulness of explainers therefore, from a practitioner's perspective (Lyu et al., 2024)

we employ our tests primarily to identify unfaithful explainers. It's important to note that our test does not take into account other criteria, such as biases in models, during the evaluation process. The scope of the work, for the time being, is restricted to NLP.

## Acknowledgement

The authors are thankful to Dr. Adolfo Escobedo, the co-author of (Yoo et al., 2020) for sharing the code for their proposed metric ($\hat{\tau}_x$). The authors also thank the reviewers for their insightful comments.

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

# 8 Appendix

## 8.1 Short Description of the Erasure Methods

We compare our findings with extensively used erasure (Jacovi and Goldberg, 2020) based metrics: comprehensiveness, sufficiency (DeYoung et al., 2019), and correlation with 'Leave-One-Out' scores (Jain and Wallace, 2019) for faithfulness comparison. Below are the definitions of these metrics.

**Comprehensiveness (↑)** This metric evaluates the extent to which an explanation captures the tokens crucial for the model's prediction. It is quantified by:

$$\text{Comprehensiveness} = f_j(x) - f_j(x \setminus r_j) \quad (2)$$

where $x$ is the input sentence, $f_j(x)$ is the model's prediction probability for class $j$, and $r_j$ is the set of tokens supporting this prediction. $x \setminus r_j$ denotes $x$ with $r_j$ tokens removed. A higher value indicates greater relevance of $r_j$ tokens.

For continuous feature attribution methods, we compute comprehensiveness multiple times, considering the top $k\%$ (from 10% to 100%, in 10% increments) of positively contributing tokens. The final score is the average across these computations.

**Sufficiency (↓)** This metric assesses whether the explanation tokens suffice for the model's prediction:

$$\text{Sufficiency} = f_j(x) - f_j(r_j) \quad (3)$$

A lower score suggests that $r_j$ tokens drive the prediction. As in comprehensiveness, we calculate the aggregate sufficiency.

**Correlation with Leave-One-Out scores (↑)** We compute Leave-One-Out (LOO) scores by iteratively omitting each token and measuring the change in model prediction. LOO scores represent individual feature importance under the *linearity assumption* (Jacovi and Goldberg, 2020). We then calculate the Kendall rank correlation coefficient $\tau$ between the explanation and LOO score:

$$\tau_{\text{loo}} = \text{corr}_{\text{Kendall}}(\text{explanation}, \text{LOO scores}) \quad (4)$$

A $\tau_{\text{loo}}$ closer to 1 indicates higher faithfulness to LOO importance. We have addressed $\tau_{\text{loo}}$ as *LOO* in Table 2.

## 8.2 Short Description of the Explainers

Local Interpretable Model-agnostic Explanations (LIME), introduced by Ribeiro et al. (2016) (Ribeiro et al., 2016), operates on the principle of local approximation. LIME generates explanations by fitting interpretable models to local regions around specific instances, providing insights into the model's behavior for individual predictions. This approach is particularly valuable for understanding non-linear models in a localized context.

SHapley Additive exPlanations (SHAP), developed by Lundberg and Lee (2017) (Lundberg and Lee, 2017), draws from cooperative game theory, specifically Shapley values (Shapley, 1951). SHAP assigns each feature an importance value for a particular prediction, ensuring a fair distribution of the model output among the input features. This method offers a unified framework that encompasses several existing feature attribution methods.

Gradient-based attribution methods leverage the model's gradients with respect to input features to quantify their importance. The simple Gradient method (Simonyan et al., 2013) computes the partial derivatives of the output with respect to each input feature, providing a first-order approximation of feature importance. However, this approach can suffer from saturation issues in deep networks.

To address these limitations, Sundararajan et al. (2017) (Sundararajan et al., 2017) proposed Integrated Gradients, which considers the integral of gradients along a straight path from a baseline to the input. This method satisfies desirable axioms such as sensitivity and implementation invariance, making it a robust choice for attribution.

Variants of these methods, namely Gradient × Input and Integrated Gradient × Input, incorporate element-wise multiplication with the input to account for feature magnitude. These approaches can provide more intuitive explanations, especially in scenarios where the input scale is significant.