# OpenReview forum: "Faithfulness and the Notion of Adversarial Sensitivity in NLP Explanations"
_EMNLP/2024/Workshop/BlackBoxNLP — BlackboxNLP 2024_

### Official Review · Reviewer_ark8 · 2024-09-06

**Overall Assessment:** 4
**Confidence:** 4

**Best Paper:**

1

**Best Paper Justification:**

N/A

**Comments Questions Suggestions And Typos:**

See summary of weakness

**Paper Summary:**

1. The paper introduces a faithfulness test to evaluate various explainers based on their sensitivity to adversarial inputs
2. They conduct tests on 6 post-hoc explainers over three text classification datasets and provide a thorough analysis of the results
3. They find perturbation-based methods are more sensitive to AE than those vanilla gradient-based methods.

**Summary Of Strengths:**

1. The paper is well-written and has a clear literature review to show the motivation of the work and the method choice.
2. The discussion of their method and experiment setup is very detail and easy to follow.
3. The experiment is quite comprehensive, and it's interesting to see how those different post-hoc explainers behave differently across different kinds of adversarial attacks and erasure-based tests. Like Gradient × Input is one of the best explainers on adversarial sensitivity, while Gradient being the worst among all.

**Summary Of Weaknesses:**

1. Overall, the paper is well-written, but having one example/figure to introduce adversarial examples and how we expect the explainer to behave would be very helpful for readers to understand this paper.
2. Simplify unnecessary mathematical notations.  For example, the author uses $\lambda$ to denote adversarial sensitivity in Definition 3, while it has never been used in later sections. The method of this paper is quite intuitive, and simplifying and merging those equations would help improve its readability.

---

### Official Review · Reviewer_VCVT · 2024-09-08

**Overall Assessment:** 4
**Confidence:** 4

**Best Paper:**

2

**Best Paper Justification:**

Very good paper.

**Comments Questions Suggestions And Typos:**

Reference for Distill BERT missing.

**Paper Summary:**

The paper introduces a faithfulness test for model explainers that measures the explainers sensitivity to adversarial inputs. The measurement of sensitivity is based on a distance measure using Pearson correlation. The paper includes experiments testing multiple explainers using BERT and Distal BERT models.

**Summary Of Strengths:**

The paper is very well written, well structured and the research is well motivated. The paper makes an important contribution to the research field by introducing the sensitivity to adversarial input as a method to evaluate explainers. The method s very interesting and to the best of my knowledge novel. The paper includes a comprehensive set of experiments with a wide variety of different explanation methods included. The experimental results show that vanilla gradient-based explainer and its derivatives are not sufficiently sensitive to adversarial inputs, but the perturbation-based methods tested in the paper are. This is an interesting and important finding.

**Summary Of Weaknesses:**

The experiments are conducted with two very similar models BERT and Distill BERT. It would have been interesting to see how the approach works with other models.

---

### Official Review · Reviewer_yqbP · 2024-09-15

**Overall Assessment:** 3
**Confidence:** 4

**Best Paper:**

1

**Best Paper Justification:**

-

**Comments Questions Suggestions And Typos:**

Statement made in line 44: Since erasure suggests also removing sets of features rather than only individual features, the claim that this is based on the assumption that the importance of tokens is linear.

It would be good to explicitly state what model f is in Definition 1 (It seems like this work assumes that f is a classification model)

In general, a more careful definition of notation as well as assumptions would help the reader to get a more concrete understanding. Like what does it mean that f(x)\ne f(x')? For this one class or is this somehow defined on class probability of all classes?

Definition 3 (Adversarial Sensitivity) of line 167 does not seem to give us a well defined notion of adversarial sensitivity. Since there is not one unique Adversarial Example (AE) but a set of possible AEs, as defined here lambda could take lots of different values. I suggest defining it as the maximum, mean or some other way of aggregating the distances over all AEs (for some similarity threshold), and then explicitly approximating it through sampling.

I do not fully agree with the definition of adversarial sensitivity. It relies on the assumption that if two input examples are similar with respect to S but produce different outcomes then the model uses different input features for the prediction. Yet, this does not need to be true. As an example consider the adversarial example having changed one token, let's say exchanging words like "elevator and lift" or "mom and mother", which are semantically very similar. The model might rely on the same word, but consider small nuances in formal/ informal language or different local accents to generate different model predictions. Yet, the correct attribution could still look exactly the same, namely assigning a high degree of importance on "elevator" and lift respectively, even though model prediction based on this input change has changed. Hence, while I agree that adversarial sensitivity is an interesting thing to evaluate and analyse, the framing of this way of evaluating explanations as faithfulness evaluation seems incorrect.

"more" missing in 183

S is nowhere explicitly defined.

The authors mention in line 318 that they consider an AE successful if the prediction confidence crosses a threshold of 70%. Yet, this seems to depend strongly on the task, definition of S, number of classes and should probably also be dependent on the prediction confidence of the model for the original example.

On the choice of similarity measure between explanations: The authors discuss that, since the input is perturbed in different ways, the length of the explanation vectors might also change. The discussion of defining the right similarity measure between original and perturbed examples then focuses entirely on the possibly unequal length of those explanation vectors. Yet, the authors seem to miss a more important point: Even if we aim to use a similarity metric between incomplete rankings, there still needs to be a 1 on 1 relationship between different entries in the vectors. I would argue, we should not search for a metric that can take vectors of different length as an input since a one-to-one correspondence of the tokens in the perturbed and original example can not be ensured (like one nth token being perturbed into two, making the explanation vector longer but also causing the attribution values corresponding to the nth+1 token now to appear at the nth+2nd position in the explanation vector). Since the goal of the distance metric is to measure how much the attribution values of certain tokens has changed, we should rather try to define something like a one-to-one correspondence, potentially including one-to-two correspondences (or n-to-m correspondences) where a semantically similar substitution has happened defining a extension of existing 1-to-1 based similarity measures for those correspondences.

Line 416 might have a typo since "infinitely" would make more sense here.

Table 1: Results should be reported until the last significant decimal, rather than up to a random decimal. Also if you are reporting up to a decimal, make sure even values with zeros as decimals will be reported that way. (like 25.00 for AG News A1 Acc in table 1)

Table 1 needs more discussion in the text. Why are attack 1 and 2 decreasing accuracy this much? Is this maybe a sign that model has overfitted too much to the data? Why is the accuracy after attack 3 nearly perfect? This seems odd to me, since further analysis is based on these models and attacks, more discussion would be in place here.

A big part of the Results and Discussion sections describes how a ranking of explainers is determined for each evaluation metric and how then a total ranking is determined. This should be part of the experimental setup section rather than the results as it interrupts the reading flow. Also, often combining several evaluation metrics into one is not desirable, since it is strongly dependent on use case which property on an explanation should be weighted

The framing of these tests as necessary conditions (which usually then would be considered to be a binary test) is not consistent with the use of the proposed tests as evaluation metrics with a score rather than a fail/pass of a necessary condition.

Evaluating comprehensiveness in steps of 10%, 20%, 30%,... can be rather coarse dependent on the input size. Usually we want compact explanations, I would suggest evaluating more on smaller explanation sizes. Explanations of size >30% or so of the input size do not seem to be very useful anymore and considering those in the calculated average for the final score might out-weight the actual relevant explanation sizes, especially for comprehensiveness since for higher explanation sizes the comprehensiveness scores will be much higher. The results shown in the table seem to support my criticism on the way the reported scores are being averaged over the top10%, 20%,... of features. Comp scores are very high overall pointing to too large explanation size. Similarly, Sufficiency scores are mostly very low, sometimes even 0 or negative. The analysis could benefit from using different evaluation steps.

**Paper Summary:**

This work tackles the difficult problem of evaluating explanation faithfulness for instance wise feature attribution explanations on classification tasks. It introduces a novel evaluation approach called adversarial sensitivity that uses adversarial examples, hence examples in the input space that are close to the original input vector, with respect to some distance metric, but generate a different prediction. The evaluation metric is based on the assumption that adversarial examples, should have a high degree of dissimilarity since a different prediction is being produced. Additionally, the authors suggest generating adversarial examples using only a selected number of perturbations to the input example to avoid out of distribution examples, which is what prior work is often accused of doing. The authors empirically compare the results of their evaluation metric for different kinds of perturbations with existing faithfulness metrics.

**Summary Of Strengths:**

The evaluation of explanation is a difficult and important task. This work tackles two important issues within evaluation of explanations: The OOD problem as well as the “biased for one explanation approach” problem.

The work seems novel, giving a new perspective to the field.

Most choices are discussed in detail.

The work is well motivated.

**Summary Of Weaknesses:**

I do not fully agree with the definition of adversarial sensitivity. It relies on the assumption that if two input examples are similar with respect to S but produce different outcomes then the model uses different input features for the prediction. Yet, this does not need to be true. See below for examples.

On the same line of argumentation (again see below for details) I think the framing of this work should change slightly, away from faithfulness evaluation, but rather define adversarial stability of explanations as a way to analyse the model.

Some notation and design choices are not defined making reproducibility of this work questionable.

Even though there is a long paragraph discussing the choice of similarity measure between different explanations, I feel like the authors are somewhat missing an important point: A similarity metric only makes sense if we know what attribution values relate to each other. For more details on this see below.

Overall, result and discussion section should could be structured better. Some of the results are not discussed sufficiently or carefully enough. Also some choices in the evaluation should be reconsidered.

---

### Decision · Program_Chairs · 2024-09-20

**Decision:**

Accept

**Comment:**

This paper presents a novel approach for evaluating explanation faithfulness through adversarial sensitivity, which utilizes adversarial examples to assess the stability of explanations. Reviewers agree that the paper is well-motivated and structured, with comprehensive experiments. However, the identified weaknesses, such as issues with paper presentation, clarity, and experimental design, should be addressed in the camera-ready version.